# Fate of telomere entanglements is dictated by the timing of anaphase midregion nuclear envelope breakdown

Rishi Kumar Nageshan[1] ✉, Raquel Ortega[2], Nevan Krogan [3,4,5] & Julia Promisel Cooper [1] ✉

Persisting replication intermediates can confer mitotic catastrophe. Loss of the fission yeast telomere protein Taz1 (ortholog of mammalian TRF1/TRF2) causes telomeric replication fork (RF) stalling and consequently, telomere entanglements that stretch between segregating mitotic chromosomes. At ≤20 °C, these entanglements fail to resolve, resulting in lethality. Rif1, a conserved DNA replication/repair protein, hinders the resolution of telomere entanglements without affecting their formation. At mitosis, local nuclear envelope (NE) breakdown occurs in the cell's midregion. Here we demonstrate that entanglement resolution occurs in the cytoplasm following this NE breakdown. However, in response to *taz1Δ* telomeric entanglements, Rif1 delays midregion NE breakdown at ≤20 °C, in turn disfavoring entanglement resolution. Moreover, Rif1 overexpression in an otherwise wild-type setting causes cold-specific NE defects and lethality, which are rescued by membrane fluidization. Hence, NE properties confer the cold-specificity of *taz1Δ* lethality, which stems from postponement of NE breakdown. We propose that such postponement promotes clearance of simple stalled RFs, but resolution of complex entanglements (involving strand invasion between nonsister telomeres) requires rapid exposure to the cytoplasm.

A successful mitosis ensures the accurate distribution of the replicated genome to each daughter cell. Each pair of sister chromatids must attach to the mitotic spindle such that the respective sisters are pulled toward opposite poles of the cell. Once chromosome segregation is complete, the spindle disassembles. Mammalian cells undergo an open mitosis in which the nuclear envelope (NE) breaks down completely at mitotic onset, allowing access of a cytoplasmic spindle to the chromosomes[1]. The fission yeast *Schizosaccharomyces pombe* was traditionally considered to undergo a closed mitosis in which the spindle forms and disassembles inside the nucleus. However, two separate instances of local NE breakdown are necessary for this mitosis. First, at the onset of mitosis, NE breakdown beneath the centrosome (called the spindle pole body, SPB) is required to enable SPB insertion into the NE, which is in turn required for formation of the intra-nuclear spindle[2–5]. Second, the NE breaks down locally in the midregion between segregating chromosomes toward the end of mitosis[6]. Recent studies have documented a sequential dismantlement of specialized nuclear pore complexes (NPCs) that localize to the anaphase midregion, and demonstrated that this NPC dismantlement triggers the local NE breakdown, which in turn triggers spindle disassembly[7]. Failure to remove NPCs leads to defects in spindle disassembly and failed karyokinesis.

[1]Department of Biochemistry and Molecular Genetics, University of Colorado Anschutz Medical Campus, Aurora, CO 80045, USA. [2]Department of Molecular Cellular and Developmental Biology, University of Colorado, Boulder, CO, USA. [3]Quantitative Biosciences Institute (QBI), University of California, San Francisco, CA 94158, USA. [4]Department of Cellular and Molecular Pharmacology, University of California, San Francisco, CA 94158, USA. [5]Gladstone Institute of Data Science and Biotechnology, J. David Gladstone Institutes, San Francisco, CA 94158, USA. ✉e-mail: rishi.nageshan@cuanschutz.edu; julia.p.cooper@cuanschutz.edu

Unresolved replication intermediates that are carried forward to mitosis manifest as chromosome entanglements that are stretched across the anaphase midregion between segregating chromosomes[8–11]. Various repair and replication factors are known to bind and act on these entanglements, but how these factors impact anaphase midregion NE breakdown, and how this NE breakdown affects the fate of entanglements, is unknown[8,12–15].

Telomeres, the ends of eukaryotic chromosomes, comprise terminal G-rich repetitive DNA sequences bound by sequence-specific DNA binding proteins and associated proteins, collectively called shelterin, which distinguish the natural ends of chromosomes from those ends produced by damage-induced chromosome breakage[16,17]. Telomeres also solve the end-replication problem by recruiting the specialized reverse transcriptase, telomerase. Among the shelterin proteins, Taz1 (ortholog of mammalian TRF1/2) specifically binds to telomeric double strand (ds) DNA and forms the foundation for shelterin[18,19]. In the absence of Taz1, telomeres are deprotected and prone to non-homologous end joining (NHEJ) mediated telomere fusions and thus dicentric chromosomes that cause mitotic catastrophe[17,20,21]. However, as NHEJ occurs only in the G1 stage of the cell cycle and actively growing fission yeast cells spend very little time in G1, they experience virtually no NHEJ[20]. This unique feature of the fission yeast cell cycle enables the investigation of additional functions of Taz1, such as its role in semi-conservative telomere replication, without the complication of chromosome end fusion-mediated cell lethality[22,23].

The repetitive G-rich nature of telomeric sequences poses a challenge for replication fork (RF) passage that is relieved by Taz1 binding[22]; mammalian TRF1 plays an analogous role in promoting fork passage[24–27]. In the absence of Taz1, the incoming replisome stalls at telomeres, and processing by sumoylated Rqh1 (the fission yeast RecQ helicase) prevents telomeric RF re-start[28]. As the telomeres at all chromosome ends are homologous and likely to sustain stalled RFs when Taz1 is absent, single strand (ss) DNA from one stalled RF can invade another such RF at a different chromosome end, forming nonsister telomere entanglements[8] (Fig. 1a). In the ensuing mitosis, these entanglements are stretched across the cellular midregion between the segregating chromosomes, and are resolved during anaphase in cells grown at 32 °C but not at temperatures ≤20 °C[29]. Hence, taz1Δ cells are cold sensitive (hereafter referred to as c/s). Telomere entanglements form independently of NHEJ factors and in the absence of Rad51, suggesting that telomere-to-telomere strand invasion is independent of standard homologous recombination pathways. Some features of telomere entanglements are apparent from live analysis of mitosis, which reveals the telomere protein Tpz1 and the ssDNA binding protein complex Replication Protein A (RPA) bound to entanglements that stretch across the anaphase midregion and are interspersed with histone signals.

The conserved multifunctional replication/repair protein Rif1 is partially responsible for hindering telomere entanglement resolution and thus conferring taz1Δ c/s[8,30]. First identified as a Rap1 interacting factor in budding yeast, Rif1 controls telomere length and silencing in budding and fission yeast[30–32]. However, as fission yeast Rif1 binds telomeres through Taz1, the role of Rif1 in opposing taz1Δ telomeric detanglement is independent of its canonical interaction with telomeres[30,32]. Crucial nontelomeric roles for mammalian and yeast Rif1 abound, including the regulation of DNA replication timing and DNA repair pathway choice[33–38]. These Rif1 functions are exerted via its interaction with Protein Phosphatase 1 (PP1) family proteins; indeed, Rif1 can be thought of partly as a scaffold that links PP1 to specific nuclear sites at specific cell cycle stages[36,39–41]. Loss of Rif1 fails to prevent RF stalling at taz1Δ telomeres; Rif1 also fails to affect the processing of stalled RFs to form telomere entanglements. Instead, the role of Rif1 in tazΔ c/s is exerted specifically during anaphase, as demonstrated using the separation-of-function S-rif1+ allele, which retains Rif1's S-phase functions but loses its mitotic functions and in doing so, rescues taz1Δ c/s[8].

To delineate the mechanism(s) by which Rif1 inhibits taz1Δ telomere detanglement, we performed a series of synthetic genetic array screens to find genes whose deletion reverts the suppression of taz1Δ c/s by rif1Δ or S-rif1+, as well as a more complete compendium of taz1Δ c/s suppressors[42]. Surprisingly, gene deletions that delay anaphase midregion NE breakdown negate the ability of Rif1 loss to rescue taz1Δ c/s. Conversely, gene deletions that advance midregion NE breakdown suppress taz1Δ c/s. Accordingly, we find that taz1Δ telomere entanglements cause a Rif1-dependent delay in midregion NE breakdown, which in turn compromises the cell's ability to resolve entanglements. Indeed, live microscopy reveals that the majority of entanglements are resolved after midregion NE breakdown, in the cytoplasm. Rif1 overexpression leads to cold-specific NE expansion and cell death, which is subverted by treating cells with a membrane fluidizing agent. These results pinpoint NE fluidity and the nuclear versus cytoplasmic milieus as arbiters of the final correction mechanisms that ensure chromatin clearance between segregating chromosomes and their successful inheritance.

## Results

### Loss of nuclear pore complex components promotes resolution of telomere entanglements

Our previous studies demonstrated that Rif1 inhibits the resolution of taz1Δ telomere entanglements at ≤20 °C without affecting telomeric RF progression[8]. A convenient marker of telomere entanglements is the appearance of aberrant bridges that stretch between the separating chromatin masses at anaphase and are bound by RPA, a marker of ssDNA (Figs. 1b and S1a, b). These aberrant RPA bridges often appear punctate and terminate asymmetrically with a concentrated globule of RPA (Fig. 1b). These appear in taz1Δ cells at both 32 °C and 19 °C (note that we routinely incubate cells at 19 °C to assess c/s), but are resolved only at 32 °C[8] (Fig. 1b; RPA panels). To deepen this analysis of telomeric entanglements, we considered that sumoylated Rqh1 acts on stalled taz1Δ telomeric RFs to prevent RF re-start[28]; a sumoylation deficient Rqh1 allele (rqh1-SM) enables RF re-start, thereby mitigating taz1Δ telomeric entanglement[28]. Hence, while taz1Δ cells with proficient rqh1+ display an increased frequency of aberrant RPA bridges, this frequency is significantly reduced in the rqh1-SM background (Figs. 1d and S1c), confirming that these RPA bridges are products of aberrant processing of stalled RFs.

Unresolved RPA bridges manifest as chromosome segregation defects in the subsequent cell cycle, in which DNA breakage and activation of DNA damage response pathways are observed[8]. Hence, a useful comparison to RPA bridge formation is provided by analysis of bulk chromatin segregation monitored via a tagged histone (Fig. 1c)[8,10]. Unlike RPA bridges, which appear in taz1Δ cells at 32 and 19 °C, bulk taz1Δ chromosome segregation defects are mainly restricted to 19 °C (Fig. 1c, e), confirming that RPA bridges are resolved at 32 °C. As we previously demonstrated[8], rif1+ deletion has little impact on RPA bridge frequency, reiterating that Rif1 affects neither telomeric RF stalling nor stalled RF processing (Fig. 1b, d). However, rif1+ deletion suppresses the chromosome segregation defects seen via tagged histones at 19 °C, indicating that the aberrant RPA bridges are completely resolved at anaphase[8] (Fig. 1c, e). Thus, Rif1 specifically inhibits the resolution, not the formation, of telomeric entanglements.

To investigate the mechanisms that underlie taz1Δ entanglement resolution, we performed a series of parallel screens in which the S. pombe gene deletion library was crossed with taz1Δ, rif1Δ and taz1Δ-rif1Δ cells to identify gene deletions that rescue taz1Δ c/s or reverse the suppression of c/s conferred by rif1+ deletion[42]. Based on our previous work, we expected several genetic interactions with taz1Δ. For instance, we know that rap1+ deletion confers synthetic growth defects with taz1Δ at 19 °C[30] while ulp2Δ suppresses taz1Δ c/s[28]. Both of

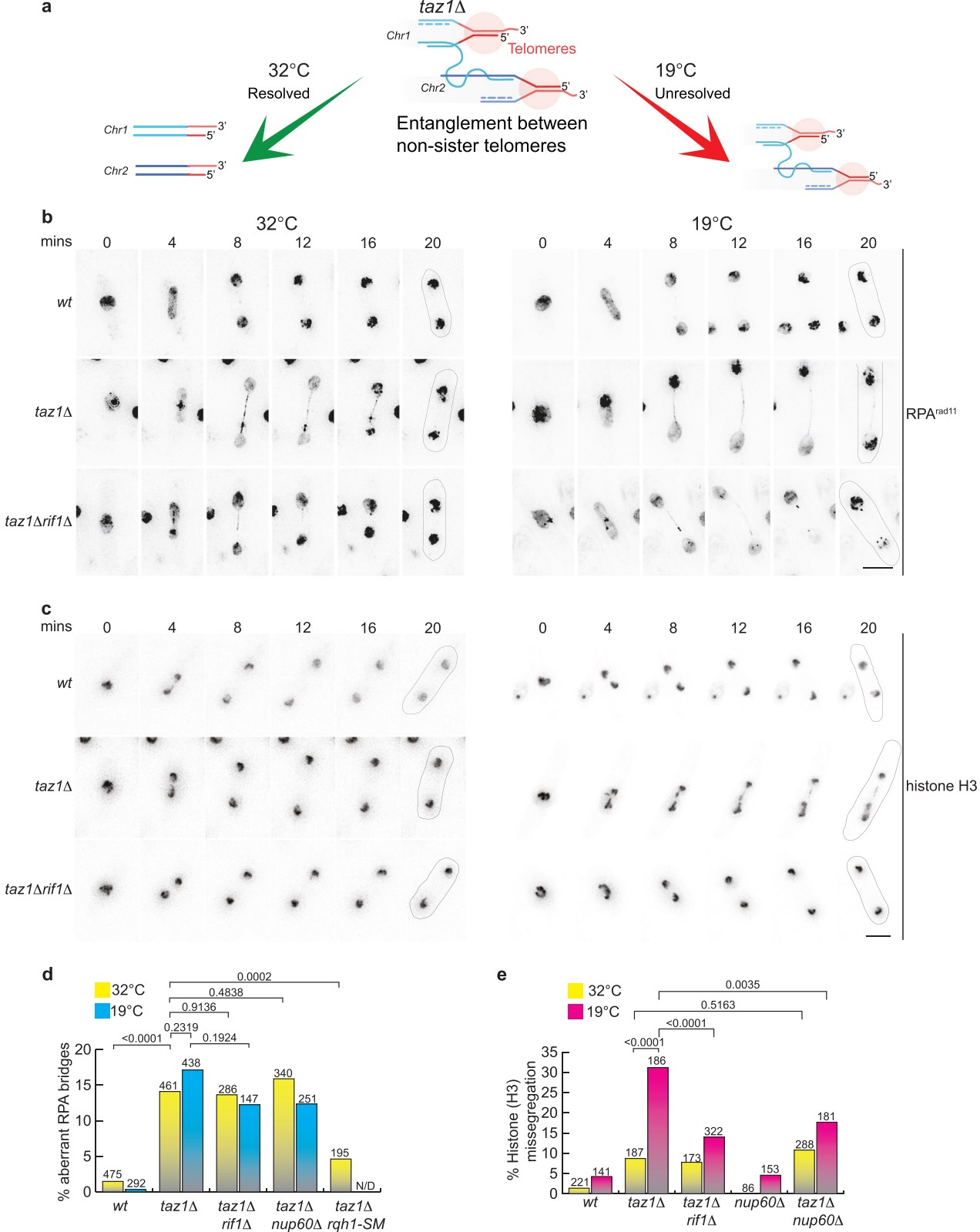

these interactions were recapitulated in our screens, validating their efficacy[42].

Deletion of the genes encoding Nup60 or Nup132, both components of the nuclear pore complex (NPC), yielded suppression of *taz1Δ* c/s (Fig. 2a). Epistasis analysis in which *taz1Δrif1Δnup60Δ* triple mutants were compared with *taz1Δnup60Δ* and *taz1Δrif1Δ* double mutants revealed similar degrees of c/s rescue in the double and triple

mutants (Fig. 2b), suggesting that Nup60 and Rif1 act via a common pathway.

If Nup60 and Rif1 act via a common pathway, *nup60Δ*, like *rif1Δ*, would confer *taz1Δ* telomere entanglement resolution without affecting entanglement formation. To investigate this, we compared RPA bridge frequencies in *taz1Δ* cells with and without *nup60+* at 32 and 19 °C. Consistent with the observed epistatic interactions, loss of

**Fig. 1 | Telomere entanglements fail to resolve in the cold. a** Representation of non-sister telomere entanglement formation, resolution at 32 °C, and failed resolution at 19 °C. **b** Frames from representative films of live mitotic cells as visualized by C-terminal tagging of the RPA subunit Rad11 with GFP at its endogenous locus. Aberrant RPA[Rad11] bridges are seen in *taz1Δ* and *taz1Δrif1Δ* cells. An ectopic copy of mCherry-α-tubulin (at the *aur1+* locus) was expressed to assess cell cycle stage (Fig. S1a, b). 0′ = metaphase; subsequent increments represent time in anaphase (Fig. S1a, b). All time lapse images henceforth are labeled identically. Cells were maintained in log phase at 32 °C (1 day) or 19 °C (3 days) before imaging. Scale bars represent 5 μm here and throughout the paper unless otherwise specified. Cell outlines determined from brightfield images are shown for each filmed cell in the last frame. **c** Frames from films of live chromosome segregation visualized via tagged histone H3 (*hht1-RFP*). Cells were maintained in log phase either at 32 °C (1 day) or 19 °C (3 days). **d** Quantitation of aberrant RPA bridges from cells grown at 32 °C (1 day) or 19 °C (3 days). *n* values shown above each bar. N/D not determined. **e** Quantitation of missegregation from films as in (**c**). Exact *p* values derived from two tailed Fisher's exact tests are represented above the brackets in (**d**) and (**e**). Source data are provided for all figures in the Source Data file.

*nup60+* suppresses *taz1Δ* c/s without affecting the formation of aberrant RPA bridges (Figs. 1d and S1d). Moreover, Nup60 loss suppresses the appearance of *taz1Δ* chromosome missegregation as observed via tagged histones (Figs. 1e and S1e), again phenocopying the loss of Rif1. This confirms that the suppression of *taz1Δ* c/s by the loss of Nup60 is not due to averted RPA bridge formation; rather, Nup60 and Rif1 act via a common pathway that regulates entanglement resolution.

### Telomere entanglements cause a Rif1 dependent delay in anaphase midregion NE breakdown

Both Nup60 and Nup132 associate with the nucleoplasmic surface of the NPC and are present along the anaphase midregion NE (Fig. 2c)[6,7], placing them in the cellular domain harboring unresolved telomere entanglements at anaphase. We also observe Rif1's presence in the midregion both in *wt* and *taz1Δ* cells (Fig. S1f, g). This Rif1 signal is barely visible in wt cells at 32 or 19 °C, or in *taz1Δ* cells at 32 °C. However, Rif1 intensely localizes to the midregion in *taz1Δ* cells maintained at 19 °C. As the removal of Nup60 and Nup132 is a key trigger for the localized anaphase midregion NE breakdown[6,7], suppression of *taz1Δ* c/s by deletion of *nup60+* or *nup132+* might reflect a role for anaphase NE breakdown in telomere entanglement resolution. To address this possibility, we monitored NPC disassembly and anaphase midregion NE breakdown. NPC disassembly (monitored via the loss of Nup60-mCherry signal) in this region is significantly delayed in *taz1Δ* cells at 19 °C but not at 32 °C (Fig. 2d, e). The delay is independent of cell size, as *taz1Δ* cells lacking Chk1, which show normal cell size due to G2/M checkpoint inactivation, show an extent of midregion NE breakdown delay similar to that of *taz1Δ chk1+* cells. Crucially, this delay depends on Rif1 (Fig. 2d, e).

To directly assess the integrity of the NE, we utilized NLS-GFP-β-Gal as a marker for nucleoplasmic content; loss of NLS-GFP-β-Gal signal indicates leakage of nuclear contents and thus loss of NE integrity. While *wt* cells show leakage of NLS-GFP-β-Gal -10 min after anaphase onset, *taz1Δ* cells show a cold-specific delay (averaging -12 min post-anaphase onset) in midregion NE breakdown (Fig. 2f, g). As seen for NPC disassembly, the delay in local NE breakdown is Rif1 dependent (Fig. 2f, g). These observations suggest that Rif1 responds to *taz1Δ* telomere entanglements by conferring a delay in anaphase midregion NE breakdown at 19 °C; in turn, this delay stymies resolution of entanglements, leading to Rif1- and Nup60/Nup132-dependent c/s (Fig. 2f, g).

### Telomere detanglement occurs in the cytoplasm

The foregoing observations suggest a model in which advanced timing of the exposure of telomere entanglements to the cytosol enables entanglement resolution. To assess directly the idea that resolution occurs only upon cytoplasmic exposure, we monitored the disappearance of the inner NE protein Bqt4 relative to entanglement resolution. To determine whether Bqt4 is a useful marker for anaphase midregion NE breakdown, we first co-filmed Bqt4 with Nup60 and spindle microtubules. Unlike NPC (Nup60) distribution, which is discontinuous across the midregion with enrichment toward the center[6] (Fig. 2d), Bqt4 covers the entire NE, forming a continuous bridge

between the dividing nuclei (Figs. 3a and S2a). At the onset of anaphase midregion NE breakdown, this bridge disassembles from the center outwards, ending at the base of each daughter nucleus. The onset of Bqt4 disassembly is temporally sandwiched between Nup60 disassembly (which occurs 20 s before Bqt4 disassembly) and onset of spindle disassembly (10 s after Bqt4 disassembly; Figs. 3b and S2a–c), which requires the local NE breakdown[6,7]. Spindle disassembly itself is coupled with a sudden movement of daughter nuclei toward each other; this movement always occurs just after Bqt4 disappears. Hence, Bqt4 is a precise and informative marker of the midregion NE and its breakdown (Fig. 3b).

With Bqt4 visualization as a tool for NE assessment, we monitored the resolution of RPA bridges in *taz1Δ* cells grown at 32 °C, a permissive temperature at which telomere entanglement resolution occurs readily. This analysis shows clearly that RPA bridge resolution occurs in the cytoplasm following local NE breakdown (Fig. 3b and Movies S1–S4); indeed, RPA bridge resolution occurs following or coincident with Bqt4 disassembly in 80% of cells (Fig. 3c). Those 14% of RPA bridges that resolve prior to anaphase midregion NE breakdown likely represent sister telomere entanglements, the products of single stalled forks that persist but do not become involved in strand invasions with non-sister chromosomes. To further probe the cytosolic nature of entanglement resolution, we utilized an independent marker, mCherry-ADEL, which localizes to the ER and all membranes derived therefrom, including the NE (Fig. S2d). Complementing our Bqt4 data, we observe discontinuous membrane staining with ADEL prior to RPA bridge resolution, indicating local NE breakdown, around the RPA bridges before their resolution, again verifying that entanglement resolution is a cytoplasmic event.

### Delayed midregion NE breakdown inhibits telomere entanglement resolution

Not only did our suppressor screen for *taz1Δ* c/s reveal anaphase midregion NE breakdown as a key player in telomere entanglement resolution, but also NE breakdown emerged from our parallel synthetic lethal screen. For instance, we observe a negative genetic interaction between *taz1+* and *imp1+* (which encodes importin α) even at permissive temperature (Fig. S3a; *taz1Δimp1Δ* cells are too inviable to study at 19 °C). As Imp1 is essential for midregion NPC disassembly[7], the midregion is hyperstabilized in *imp1Δ* cells (Fig. S3b). As we find a Rif1-dependent delay in midregion NE breakdown only at cold temperature, it appears that *imp1+* deletion mimics some feature of the 19 °C condition (Fig. S3c–d)—below we present data suggesting that this feature is increased NE rigidity.

A second independent line of evidence pointing to a regulatory role for anaphase midregion NE breakdown in telomere entanglement resolution comes from our screen for gene deletions that avert the suppression of *taz1Δ* c/s afforded by *rif1+* deletion. Deletion of the gene encoding Mto1 (microtubule organizer 1) yields a particularly strong synthetic sickness with either *taz1Δrif1Δ* or *taz1ΔS-rif1*[42]. This genetic interaction was validated by constructing new *mto1+* deletions in independently generated strains of each relevant genotype (Fig. 4a); *mto1+* deletion has no impact on c/s in the respective single mutant backgrounds. Furthermore, loss of *mto1+* in a *taz1Δrif1Δ* background

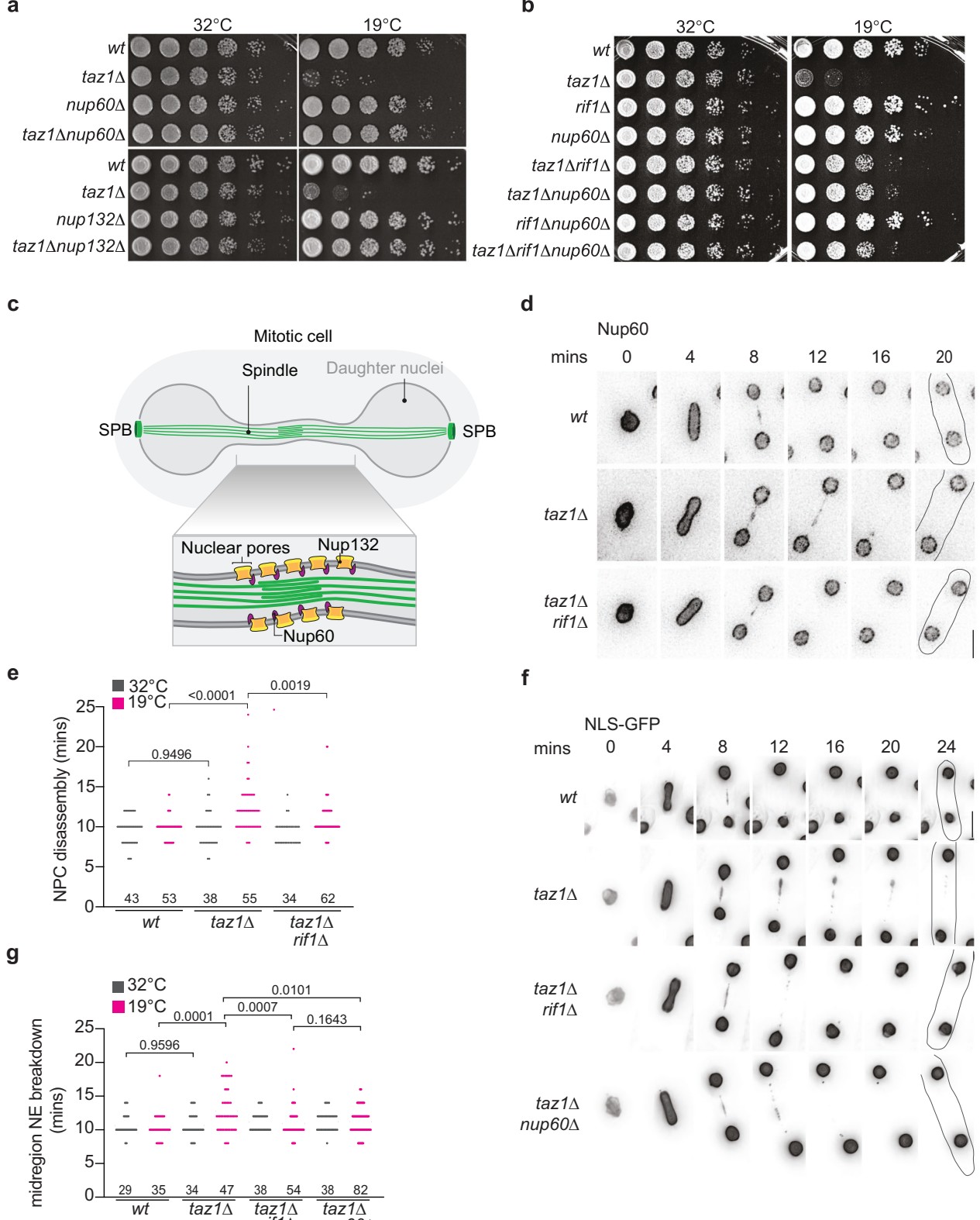

**Fig. 2 | Rif1 causes NPC dependent delay in anaphase midregion NE breakdown in *taz1Δ* cells. a** Five-fold serial dilutions of log phase (32 °C) cells were incubated at 32 °C (2 days) or 19 °C (7 days). **b** Five-fold serial dilution as in (**a**). **c** Schematic of mitotic cell showing the anaphase midregion harboring nuclear pore complexes, which disassemble at anaphase, leading to local NE breakdown. **d** Frames from films of mitotically dividing cells of represented genotypes expressing Nup60-mCherry grown at 19 °C. **e** The timing of NPC disassembly is plotted relative to anaphase onset from cells maintained either at 32 or 19 °C. **f** Frames from films of mitotically dividing cells at 19 °C expressing NLS-GFP-βGAL (a marker of nuclear content) under control of the *nmt1+* promoter; cells were pre-grown in PMG media without thiamine to induce expression. **g** Quantitation of midregion NLS-GFP-βGAL signal loss at the indicated time points post-anaphase onset from cells maintained at 32 or 19 °C. Exact *p* values derived from two-tailed Mann–Whitney test are represented above the brackets in (**e**) and (**g**). *n* values (number of cells) for each condition are provided above the respective genotypes in (**e**) and (**g**).

**a**

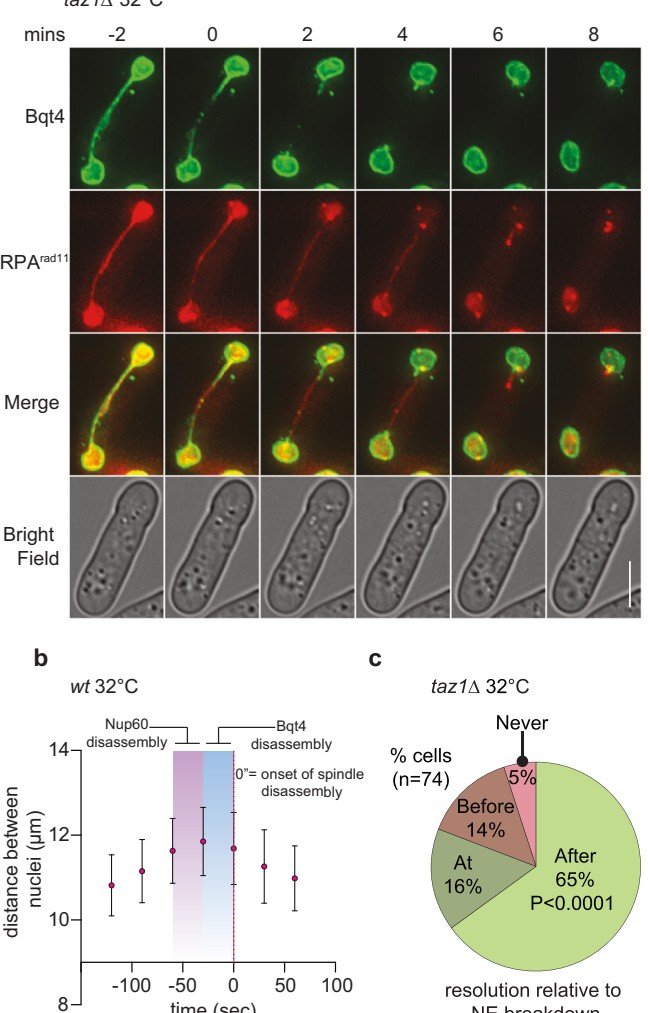

**a**

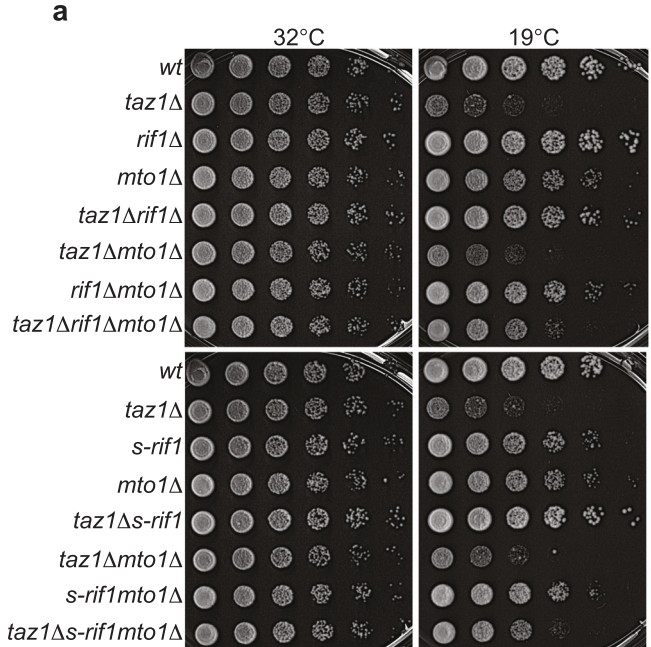

**b**

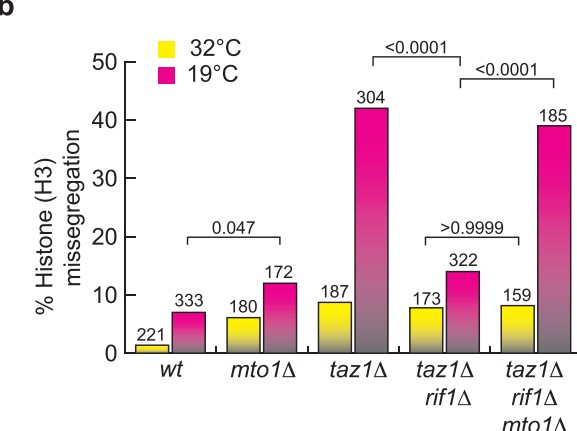

**Fig. 3 | Exposure of entanglements to cytoplasm is required for the resolution.**
**a** Frames from films of mitotically dividing *taz1Δ* cells maintained in log phase at 32 °C. These cells expressed an additional copy of GFP-Bqt4 from its cognate promoter integrated at the *lys1+* locus, while endogenous *bqt4+* remained unmodified. Rad11 is C-terminally tagged with mCherry at its endogenous locus. Time 0′ marks the onset of Bqt4 disassembly, i.e. the onset of anaphase midregion NE breakdown. **b** Graph of relative time of Nup60 and Bqt4 disassembly from the anaphase midregion relative to the distance between daughter nuclei, whose sudden decrease occurs at the onset of spindle disassembly. Mean ± standard deviation is represented from 23 cells analyzed. **c** Quantitation of entanglement resolution from films as in (**a**). Distribution of percentages of cells that resolved entanglements at various times relative to anaphase midregion NE breakdown. Exact *p* values derived from two tailed Fisher's exact *T* test are represented.

**c**

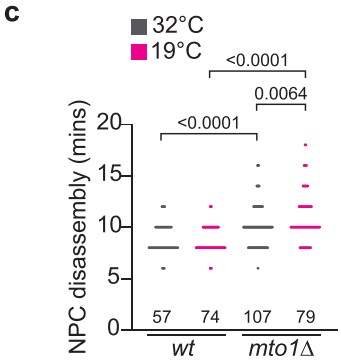

hinders resolution of telomere entanglements as evinced by increased levels of chromosome missegregation in the triple mutant cells relative to the double mutant (Fig. 4b), reverting chromosome missegregation levels to those seen in unsuppressed *taz1Δ* cells at 19 °C.

Mto1 is a component of cytoplasmic microtubule organizing centers and is required for the nucleation of microtubules; it nucleates cytosolic microtubules during interphase, and equatorial and astral microtubules during mitosis[43,44]. Mto1 also localizes to the SPB during interphase, when the SPB resides on the cytoplasmic surface of the NE; this localization is required for interphase SPB-microtubule interactions[43]. Mto1 mutant alleles that are deficient in forming

**Fig. 4 | Delayed cytoplasm exposure hinders telomere disentanglement. a** Five-fold serial dilutions of cells grown to log phase at 32 °C were incubated at 32 °C (2 days) or 19 °C (7 days). **b** Quantitation of cells showing aberrant histone-H3 segregation when grown at 32 °C for 1 day or 19 °C for 3 days (as in Fig. 1e). For comparison, data from *wt* and *taz1Δ* cells at 32 °C and *taz1Δrif1Δ* cells at 19 °C and 32 °C are re-plotted from Fig. 1e. Exact *p* values determined from two tailed Fisher's exact *T* test are represented above the brackets. *n* values are above each bar. **c** Quantitation of the timing of NPC disassembly time post-anaphase onset in cells maintained either at 32 °C for 1 day or 19 °C for 3 days. Exact *p* values from two-tailed Mann–Whitney test are indicated above brackets. *n* values are above each genotype.

either equatorial (*mto1-427*) or astral microtubules (*mto1-(1-1065)*) fail to phenocopy *mto1Δ* in a *taz1Δrif1Δ* setting (Fig. S3e, f). Thus, neither disruption of equatorial nor astral microtubules alone is sufficient to confer restoration of *taz1Δrif1Δ* c/s[45]. Moreover, Mto1-(1-1065) fails to localize to the interphase SPB, suggesting that interphase microtubule-SPB interactions are not crucial to *taz1Δ* telomere entanglement resolution.

While Mto1 is dispensable for formation and elongation of mitotic spindles, *mto1Δ* cells show a delay in spindle disassembly at the end of mitosis (see below)[43]. Therefore, we investigated whether anaphase midregion NE breakdown is delayed in the absence of Mto1. Indeed, *mto1Δ* cells show a delay in anaphase midregion NPC disassembly at 32 °C and a more severe delay at 19 °C (Fig. 4c). This effect is specific to the *mto1+* null; the *mto1-427* and *mto1(1-1065)* alleles fail to confer a delay in NPC disassembly (Fig. S3f). Furthermore, loss of *rif1+* does not impact the delayed NPC disassembly observed in *mto1Δ* cells. Hence, the effect of Mto1 loss in reversing suppression of *taz1Δ* c/s by *rif1+* deletion likely stems from delayed anaphase midregion NE breakdown.

While the *mto1-427* and *mto1(1-1065)* alleles have lost equatorial or astral microtubules, they still possess the post anaphase array[45], which is lost in *mto1Δ* cells. Indeed, in *wt* cells, we observe networks of cytosolic microtubules crossing the midregion NE bridge at the time of NE breakdown at both 32 and 19 °C; these are absent in cells lacking Mto1 (Fig. S4a, b). Hence, Mto1's role in NE breakdown likely involves its nucleation of these NE-prodding microtubules, which disrupt the midregion NE and promote its breakdown.

While *mto1Δ* reverts the suppression of *taz1Δ* c/s conferred by loss of Rif1, *mto1Δ* does not revert the suppression of *taz1Δ* c/s conferred by loss of Nup60 (Fig. S4c). Indeed, loss of Nup60 destabilizes the ana-phase midregion NE even in a *mto1Δ* background, as *nup60Δmto1Δ* double mutants display *wt* midregion NE breakdown timing (Fig. S4d) while *rif1Δmto1Δ* cells are delayed. Similarly, loss of *mto1+* does not affect midregion NE breakdown timing of *taz1Δnup60Δ* cells (Fig. S4d). We suggest that the loss of Nup60 guarantees a fragile midregion NE that can break down even in the absence of Mto1-nucleated micro-tubule prodding.

A conspicuous phenotype of *mto1Δ* cells is spindle persistence. It has been suggested that the lack of cytosolic microtubules in the absence of Mto1 causes an increased concentration of free tubulin, leading to net microtubule polymerization and delaying spindle disassembly[43]. Hence, we considered the possibility that spindle per-sistence, rather than midregion NE breakdown, is responsible for the inhibition of telomere entanglement resolution. However, loss of Nup60 in a *mto1Δ* background does not rescue spindle persistence even though it rescues *taz1Δ* c/s (Fig. S4e). Therefore, c/s rescue stems from the hastened midregion NE breakdown seen in *taz1Δ-nup60Δmto1Δ* (Fig. S4d), not from spindle persistence. Collectively, these results suggest a model in which Rif1 loss promotes timely NPC disassembly, advancing NE breakdown and cytoplasmic exposure of *taz1Δ* telomere entanglements; the loss of NE-prodding MTs in a *mto1Δ* setting reverses this advancement in NE breakdown, hindering entanglement resolution.

### Excess Rif1 causes NE expansion
The anaphase midregion is a well demarcated microdomain that is spatially and temporally restricted[6]. We wondered whether amplifica-tion of the effects of Rif1 throughout the nucleus would result in pan-NE defects; therefore, we overexpressed GFP-Rif1 in an otherwise *wt* setting (Fig. S5a). Rif1 overexpression leads to severe c/s (Figs. 5a and S5a, b). This c/s is likely attributable to Rif1's actions in mitosis, as *S-rif1+* cells, which overexpress Rif1 by ~3-fold in S-phase despite showing 2-fold under-expression in mitosis, grow normally at 19 °C[8]. This contrasts with the hypersensitivity of Rif1-overexpressing cells to hydroxyurea treatment, which is shared with the *S-rif1+* allele (Fig. S5c) and likely stems from exaggerating the inhibitory function of

Rif1 in restraining early origin activation, resulting in genomic under-replication. Our observations are consistent with a recent report that shows ectopic overexpression of Rif1 from a high copy number plas-mid is toxic for *S. pombe* cells[46].

To investigate the effects of Rif1 overexpression on the NE, we assessed interphase nuclear morphology via tagged inner-NE and NPC components (Man1-tomato and Nup60-mCherry, respectively). In *wt* cells grown at 19 °C, the interphase nucleus is spherical, with Man1 and Nup60 signals evenly spread around the NE (Fig. S5d). In contrast, Rif1 overexpression induces NE expansion, with folds of NE appearing and an uneven distribution of Man1 and Nup60 (Fig. S5d). This NE expan-sion is specific to Rif1-overexpressing cells grown at 19 °C; no such irregularities are seen at 32 °C (Fig. S5d). We also observe increased spindle persistence in Rif1-overexpressing cells grown at 19 °C result-ing from delayed midregion NE breakdown (Fig. S5e).

The NE expansion seen upon Rif1 overexpression demonstrates that Rif1 alters NE properties in a cold-specific manner. To explore this idea further, we treated Rif1-overexpressing cells with the membrane fluidizing agent benzyl alcohol (BA). Fission yeast cells are sensitive to BA at ≤20 °C, possibly due to slower metabolism of BA at cold tem-perature, and this is further exacerbated on the minimal medium needed for overexpression of Rif1; to circumvent this lethality, we grew the cells at moderately cold temperature (25 °C) where Rif1 over-expression is also near-lethal (Fig. 5a). Remarkably, BA treatment res-cues the severe c/s induced by Rif1 overexpression (Fig. 5a). Thus, Rif1 appears to counteract membrane fluidity, which is required for mid-region NE breakdown and in turn, telomere entanglement resolution.

### Membrane fluidizing agent suppresses *taz1Δ* c/s
As BA treatment abates the c/s stemming from Rif1 overexpression, we queried whether it could also rescue *taz1Δ* c/s by opposing the effects of Rif1 on NE rigidity. *Wt* cells lose viability progressively with increasing BA concentration at 19 °C, while no such viability reduction is seen with increasing BA concentration in *taz1Δ* cells (Fig. 5b); in contrast, BA has no effect on growth at 32 °C of either *wt* or *taz1Δ* cells. The near-lethality of BA at ≤20 °C made it impossible to confidently interpret dilution assays at this temperature. However, we were able to maintain *wt* and *taz1Δ* cells in log phase growth in the presence or absence of low concentrations of BA at 19 °C for 3 days, after which we checked the percentage of dead cells using Erythrosin B, a dye that stains dead cells. While the percentage of dead *wt* cells is unaffected by BA in liquid culture, the appearance of dead *taz1Δ* cells is partially suppressed by BA (Fig. 5c). Likewise, *taz1Δ* cells that are continuously propagated in low concentrations of BA at 19 °C show improved via-bility relative to the same cells without BA (Fig. S5f). This suppression is incomplete, likely due to the overall impact of BA on all cellular membranes. Furthermore, BA addition potentiates suppression of *taz1Δ* c/s by *rif1Δ* (Fig. 5b), enhancing suppression to the extent of recapitulating *wt* levels of growth. These data establish a key connec-tion between membrane dynamics and the temperature specificity of *taz1Δ* telomere detanglement defects. We propose that NE fluidity at 32 °C is optimal for anaphase midregion NE breakdown regardless of Rif1 status, thereby enabling *taz1Δ* entanglement resolution. However, increased membrane rigidity at 19 °C, in combination with the Rif1-promoted inhibition of NPC disassembly, causes a delay in midregion NE breakdown, which in turn impedes telomere entanglement resolution.

## Discussion
The compartmentalization of the eukaryotic genome within the nucleus prevents exposure of chromosomes to the cytosol through most of the cell cycle. Nonetheless, the limited or complete breakdown of the NE at specific cell cycle stages violates this compartmentaliza-tion. Here we find that the resulting exposure of chromatin to the cytoplasm has profound implications for resolving entanglements that

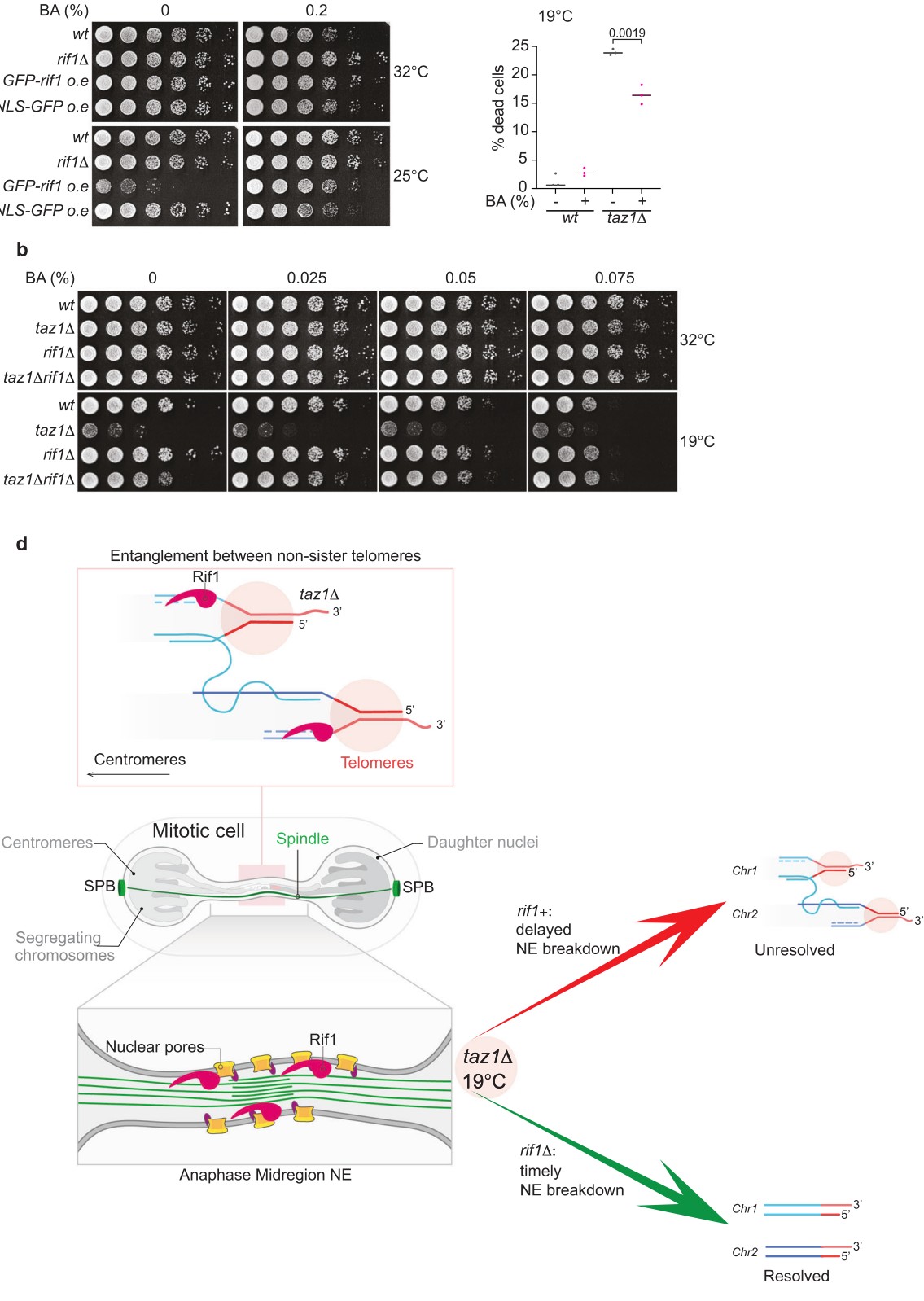

occur due to telomeric replication impediments. In response to the presence of *taz1Δ* telomeric entanglements, Rif1 mediates a delay in midregion NE breakdown, thereby prolonging the period during which DNA processing pathways operate without cytoplasmic exposure. This prolongation hinders the resolution of telomere entanglements. Our data show that Rif1 counteracts NE breakdown by altering the behavior of subunits of the NPC. The effects of Rif1 on NE biology shed light on

the reasons for the cold-specificity of the lethality caused by Taz1 loss, tying this lethality to the effects of temperature on the biophysical properties of the NE.

## Numerous genetic interactions point to NE breakdown
The convergence of numerous genetic interactions between NE breakdown-associated factors and *taz1+* deletion outlines an intimate

**Fig. 5 | Anaphase Rif1 modulates NE disassembly. a** Five-fold serial dilutions of log phase cells were plated on minimal PMG media (without thiamine to allow induction of *nmt41-gfp-rif1+*, integrated at the endogenous *rif1+* locus) with or without BA and incubated at 32 °C (3 days) or 25 °C (7 days). **b** Five-fold serial dilutions of log phase cells at 32 °C were stamped on rich media (YES) containing various concentrations of BA and incubated at 32 °C (3 days) or 19 °C (10 days). **c** Quantitation of % of dead cells grown in media without or with 0.025% BA at 19 °C for 3 days. Each data point represents one experiment in which >1000 cells were scored. The mean value from three independent experiments is represented for each condition. Exact

*p* values derived from two tailed unpaired *T* test are represented. **d** Working model for Rif1's control of anaphase midregion NE breakdown. Rif1 bound telomere entanglements (top) are shown stretched across the anaphase midregion (middle panel). The lowest panel shows Rif1 coordinating the presence of telomere entanglements with modifications to NE components including Nup60. The arrows point to the different fates of entanglements, determined by the presence or absence of Rif1. Rif1 induces a delay in exposure of entanglements to the cytoplasm, hampering resolution; conversely, timely cytoplasmic exposure, afforded by the absence of Rif1, promotes resolution.

relationship between anaphase midregion NE breakdown and telomere detanglement. The loss of NPC components whose disappearance is known to trigger localized anaphase midregion NE breakdown, Nup60 or Nup132, phenocopies and is epistatic with the loss of Rif1 in suppressing *taz1Δ* c/s. Accordingly, *taz1+* deletion leads to a cold-specific, Rif1- and Nup60/Nup132-dependent delay in anaphase midregion NE breakdown.

Our parallel screen for genes whose deletion reverses *rif1Δ*-mediated suppression of *taz1Δ* c/s also converged upon anaphase midregion NE breakdown by revealing that Mto1 is required for suppression by *rif1Δ*. Cells lacking Mto1 both show abnormal spindle persistence and recapitulate the delayed anaphase midregion NE breakdown seen in *rif1+* cells. However, as deletion of *nup60+* in the *mto1Δ* background reverses the NE breakdown delay without reversing spindle persistence, we can separate the two processes and pinpoint the former in controlling *taz1Δ* c/s. Intriguingly, these observations indicate that spindle disassembly is not automatically triggered by cytosolic exposure if free tubulin concentrations are elevated. We speculate that in addition to controlling cellular concentrations of free tubulin, the cytoplasmic microtubules nucleated by Mto1 stimulate midregion NE breakdown by 'poking' the midregion NE (Fig. S4a, b). Thus, spindle persistence and quashed resolution of telomere entanglements appear to be two independent phenomena caused by delayed anaphase NE breakdown.

### Confounded replication re-start pathways at *taz1Δ* telomeres lead to perilous anaphase

Interactions between stalled RFs and NPCs in earlier stages of the cell cycle have been implicated in RF restart. In fission yeast, stalled RFs accumulate SUMO-modified proteins, which confer Rad51-dependent relocation to NPCs where processing by the SUMO protease Ulp1 allows fork restart[47]. Likewise, S-phase localization to NPCs of stalled telomeric RFs in human cells, or of stalled RFs at CAG repeat regions in budding yeast, promotes RF re-start[48,49]. However, unlike RFs that re-start upon association with NPCs, Rad51 is dispensable for the processing of stalled *taz1Δ* telomeric RFs[29]. Moreover, while telomeres are constitutively positioned at the NE in a Taz1-independent manner, compromised localization of telomeres to the NE has no impact on *taz1Δ* c/s[28,50], suggesting that NPC-driven RF re-start is inactive at *taz1Δ* telomeres. Also of note are our previous observations that sumoylated Rqh1 promotes the aberrant processing of stalled telomeric RFs that prohibits RF re-start, suggesting that in the absence of Taz1, local sumoylation is detrimental rather than beneficial to the completion of chromosomal replication. Conceivably, such RF re-start pathways are specifically detrimental by promoting strand invasion from one stalled telomeric RF into a stalled RF on a non-sister telomere (Figs. 1a and 5d).

### Crucial connections between membrane dynamics and DNA processing events

The involvement of anaphase midregion NE breakdown in *taz1Δ* c/s, the suppression of *taz1Δ* c/s by a membrane fluidizing agent, and the cold-specific NE distortions and NE rigidity-associated lethality triggered by Rif1 overexpression, all reveal a link between the cold-specificity of *taz1Δ* lethality and NE dynamics. The Rif1-mediated delay in NE breakdown in response to telomere entanglements is itself cold-

specific. Hence, *taz1Δ* c/s provides a rare window into the profound effects of delayed cytoplasmic exposure on DNA processing events. Temperatures of ≤20 °C are commonly found in the natural habitat of fission yeast and indeed may represent physiological conditions. Moreover, depending on specific NE compositions and NPC modification states across different eukaryotic cell types and growth conditions, Rif1 may also regulate mitotic NE dynamics at temperatures greater than 20 °C. Indeed, cytoplasmic exposure of missegregating DNA is crucial in defining the outcome of mitosis in mammalian cells[51–55]; hence, principles of cytoplasmic entanglement resolution defined in fission yeast may be relevant throughout Eukarya.

Recruitment of Rif1 to the anaphase midregion could be via NE and/or DNA interactions. The N-terminus of Rif1 binds ss-dsDNA junctions[56], a likely mode of recruitment of Rif1 to *taz1Δ* telomeric entanglements (Fig. 5d, upper panel); Rif1's G-quartet binding capacity may also contribute[35,57]. However, Rif1 may also associate directly with the NE as suggested from previous studies[32–34,56,58–60]. Intriguingly, anaphase midregion localization of Rif1 is also widely conserved, as first observed in human cells[61]. Moreover, midbody-bound Rif1 has been shown to regulate the timing of abscission, the final step of cytokinesis, in an anaphase DNA bridge-independent manner[62]. Hence, both NE and DNA association likely concentrate Rif1 at the anaphase midregion, where it regulates NE dynamics, at least in part by regulating PP1-mediated dephosphorylation of NPC components and/or NE phospholipids[8] (Fig. 4d, lower panel). Indeed, a phosphoproteomic analysis of *wt*, *rif1Δ* and *S-rif1+* cells grown at 32 °C versus 19 °C identified Nup60 as a Rif1-targeted phosphatase substrate.

Human Rif1 was shown to bind non-telomeric ultrafine anaphase bridges (UFBs) and promote their resolution[15]. This positive role for Rif1 in UFB resolution might initially appear paradoxical given our observations that fission yeast Rif1 opposes the resolution of telomeric entanglements. However, we surmise that the Rif1-mediated delay of NE breakdown has evolved to promote resolution of sister entanglements, the most common type of chromosome entanglement arising from incomplete replication and the likely scenario in cells displaying non-telomeric UFBs. Only in rarer scenarios do non-sister entanglements arise, including the *taz1Δ* scenario in which multiple telomeres at different chromosome ends will harbor stalled RFs, providing multiple substrates for non-sister strand invasions and the formation of nonsister entanglements, whose resolution is opposed by Rif1[8].

The crucial question raised by our work is how timely cytosolic exposure promotes resolution of *taz1Δ* telomeric entanglements. NE breakdown may expose entanglements to cytosolic factors that promote resolution. Alternatively, exposure to the cytoplasm may change or halt the activities of nuclear DNA processing factors that control telomere detanglement. A prime candidate for such regulation is topoisomerase 2 (Top2), which plays an important role in *taz1Δ* telomere entanglement resolution[29,63]; current experiments aim to address the idea that Top2 activity changes upon exposure to the cytoplasm.

### Conserved mechanisms of DNA processing across a continuum of extents of NE breakdown

While a continuum of the extent of NE breakdown exists across eukaryotes, the closing stages of mitosis are universally followed by the formation of daughter nuclei that must each be encapsulated in a

new membrane. In human cells that undergo complete mitotic NE breakdown, telomere fusions that elicit dicentric chromosomes lead to the formation of NE bridges between post-mitotic daughter cells. These bridges persist into the next interphase, when bridge rupture leads to untimely cytoplasmic exposure with profound consequences for chromosome segregation, including chromothripsis and kataegis. Moreover, telomeric DNA and RNA released into the cytoplasm lead to autophagy and activation of innate immune signaling[51–53,55,64,65]. Our work demonstrates that different types of telomeric associations trigger different resolution processes. We suspect that mammalian telomere entanglements analogous to the non-sister entanglements we study herein form upon telomeric replication stress, and that Rif1 influences whether the resulting telomeric anaphase bridges become encapsulated in new NEs and whether such NEs harbor NPC components[54]. It will also be crucial to determine whether the DNA processing events that govern mammalian DNA entanglement resolution are similarly regulated by cytosolic exposure across species.

## Methods

### Materials
Media comprised YES broth (Sunrise #2011-500), YES agar (Sunrise #2012-500), and PMG (Sunrise #2060-500). Precast gels (Mini-PRO-TEAN TGX Stain-Free Gels #4568084) were used with Biorad buffer (#1610732 and #1610734). Media and growth conditions were as previously described[66]. Strains are listed in Table S1 in the Supplementary Information file. Oligonucleotide sequences are listed in Supplementary Data 1.

### Strain construction
Gene deletions and tag insertions were generated as described previously[67]; further strains were created through crossing, sporulation and selection. Tagging was confirmed by sequencing the tag-gene junctions. Diploids with heterozygous gene deletions were created either by gene deletion in preselected diploids or crossing two haploids followed by diploid selection via *ade6-210/216* complementation.

### Cell growth
All experiments were performed on logarithmically growing cells, which were grown in either YES or PMG with supplements, from a single colony at the indicated temperature. The log phase cells (0.5 OD) were diluted to 0.08–0.1 OD, incubated at the indicated temperatures and maintained in log phase by dilution with fresh media every day. This approach was used for microscopy and western blot analysis.

### Benzyl alcohol (BA) treatment
Cultures were grown as described above to saturation, diluted to 0.1 OD in YES broth without BA and incubated at 32 °C. Once the cultures reached OD 0.5, they were diluted to 0.1 OD in YES broth with varying concentrations of BA (0, 0.0025%, 0.05 or 0.075%) and transferred to 19 °C. The media was replaced daily with fresh broth containing the respective concentrations of BA. After 3 days, the cells were stained with Erythrosine B, imaged and counted using a hemocytometer; 300 cells were plated on YES media without BA and incubated at 32 °C for 2 days.

### Microscopy
Mid-log-phase cells were adhered to 35-mm glass culture dishes (Mat-Tek Corporation) using 0.2 mg/ml of soybean lectin (Sigma-Aldrich) and filled with PMG broth. Time lapse images were captured in an environmental chamber set at 32 °C with a DeltaVision Ultra with a 60x/1.42 (lens ID 10612) objective with 1.585 oil immersion, and a charge coupled device EDGE sCMOS camera. Cells were imaged every 2 min for 40 min over 16 focal planes of 0.4 µm. These acquired images were deconvolved (conserved ratio method) and quick-projected into a 2D image using the maximum intensity setting in SoftWorx (GE Healthcare Life

Sciences, versions 7.2.1 and 7.2.2). Images using the DeltaVision OMX Flex (in conventional mode) were acquired and processed similarly.

### Protein extraction and western blot
Fifty ml of log phase cultures (OD of 0.5) were pelleted and washed three times with autoclaved water before protein extraction. The cells were ruptured using a beadbeater with 1.9 g of glass beads in SDS buffer (1X phosphate buffered saline, 10% Glycerol, 2 mM EDTA, 2 mM DTT, Roche protease inhibitor cocktail, 2 mM PMSF, 1% SDS)[68]. As indicated, cells were treated with 20% trichloroacetic acid and neutralized with 1 M tris base before harvesting in SDS buffer.

For western blot analysis, 100 µg total protein was resolved on a BioRad stain-free precast gel and transferred onto a PVDF membrane using tris-glycine buffer with 20% methanol. The membrane was blocked with 5% non-fat milk and washed with PBS containing 0.05% Tween 20. Membranes were incubated in primary antibody (α-GFP antibody, ChromoTek, H39, 1:1000 dilution in blocking buffer) followed by HRP conjugated secondary antibody (α-Rat-HRP conjugate goat antibody, vector labs, PI-9400, 1:10,000 dilution in PBST), and developed in a BioRad Imager using ECL reagent (Amersham; RPN2236).

### Spindle analysis
Mitotic cell progression was monitored by observing spindle dynamics as described in Fig. S1a, b. In an unperturbed cell, the metaphase spindle is maintained at <3 µm length and elongates exponentially during anaphase.

### Statistical analysis
Statistical tests were performed in GraphPad Prism (GraphPad Software Inc., version 10.1.0). Non-parametric analysis using the Mann–Whitney test method was performed to analyze anaphase midregion NE breakdown, NPC disassembly, % dead cells, viability, and spindle persistence experiments. A two tailed Fisher's exact *T* test was performed to determine significance when analyzing percent RPA bridges and histone segregation. Double blind experiments were performed for Fig. S4d. Randomly selected two independent biological clones were analyzed for most experiments. The only exceptions were in cases for which we have captured tagged histone H3 data many times and found no variation between experiments (this is the case for *wt*, *taz1Δ* or *rif1Δ* single mutants)[8]; we used only one biological replicate of only those well-characterized genotypes (this helped in instances where many samples were filmed on a single day along side with *wt* controls). Wt control cells were always run in parallel to all queried genotypes. In experiments monitoring RPA[rad11] bridges, NPC disassembly and midregion NE breakdown (via NLS-GFP-βgal), two independent clones were imaged and analyzed. Data acquired from different clones across different dates show no significant differences; hence, such data has been pooled and represented. At least two independent clones were used to verify all the novel genetic interactions in the growth-based experiments; one clone is represented in the figures. All attempts to reproduce experimental results were successful. No specific statistical methods were employed to predetermine *n*-values.

### Image processing
All the images for a given experiment are acquired at the same microscopic settings. The quick projected images are represented in all figures except for Fig. S5d, in which a single Z stack with maximum intensity is represented. The projected images were processed in ImageJ software (open source, versions 2.1.0/1.53c); brightness and contrast were set similarly to the corresponding *wt* cells, saved as.tif files, inverted and the resulting images are represented in the figure panels.

To represent the weak signal of Rif1-GFP, the contrast is enhanced by the ImageJ software using the "enhanced contrast" function at 0.3%.

Z-stacks that comprise the Rif1-GFP (Fig. S1f) and Nup60-mCherry (Fig. S1g) signals were projected in the figures.

## Reporting summary

Further information on research design is available in the Nature Portfolio Reporting Summary linked to this article.

## Data availability

Strains and any further information are available on request. Source data are provided with this paper.

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

## Acknowledgements

We thank our lab members, all of whom have contributed vital ideas and advice, Lakshmi Sreekumar for help in double blind experiments (Fig. S4d), Michael Lichten (NCI), Dermot Cooper (University of Cambridge), and Eros Lazzerini Denchi and his lab (NCI) for crucial discussions. We thank Ken Sawin (Wellcome Center, Edinburgh) for sharing *mto1+* separation of function mutants. We particularly thank Maria Diaz de la Loza for help with illustrations. This work was supported by the National Cancer Institute intramural program and the University of Colorado School of Medicine.

## Author contributions

R.K.N. performed all the experiments, with help from R.O. in strain construction and data analysis. N.K. hosted R.K.N. in his lab to perform the synthetic genetic array screens. R.K.N. and J.P.C. conceived the study, designed and interpreted the experiments, and wrote the manuscript.

## Competing interests

The Krogan Laboratory has received research support from Vir Biotechnology, F. Hoffmann-La Roche, and Rezo Therapeutics. N.K. has a financially compensated consulting agreement with Maze Therapeutics. N.K. is the President and is on the Board of Directors of Rezo Therapeutics, and he is a shareholder in Tenaya Therapeutics, Maze Therapeutics, Rezo Therapeutics, and Interline Therapeutics. The remaining authors declare no competing interests.
