## [Peer Review File · Nature Communications]

Fate of telomere entanglements is dictated by the timing of anaphase midregion nuclear envelope breakdownEditorial Note: This manuscript has been previously reviewed at another journal that is not operating a transparent peer review scheme. This document only contains reviewer comments and rebuttal letters for versions considered at *Nature Communications*.

REVIEWERS' COMMENTS

Reviewer #1 (Remarks to the Author):

It was a pleasure to read this revised manuscript, and the manuscript, already of high quality at the start, is even better now. In particular, I appreciate the inclusion of a new Figure 1 that will help orient readers and set the stage for the rest of the study.

I believe that all my concerns have been addressed satisfactorily, and to the extent that I am able to evaluate this, the comments of the other reviewers as well. I recommend publication of the study.

Reviewer #2 (Remarks to the Author):

I am happy with the answers provided to my initial concerns and I deem the present manuscript suitable for publication in Nature Communications

Reviewer #3 (Remarks to the Author):

My concerns have been addressed by the revisions. I support publication in Nature Communications.

Reviewer #4 (Remarks to the Author):

The authors have added an admirable number of additional experiments to support the model that Rif1 impacts NE properties in a way that contributes to NE breakdown and telomere disentanglement. I appreciate the thoughtfulness with which they addressed all of my concerns regarding their prior conclusions. I'm very supportive of publishing this work and look forward to seeing it in print.